# Clinical Utility of Genomic Assay in Node-Positive Early-Stage Breast Cancer

**Mehrnoosh Pauls * and Stephen Chia ***

Department of Medical Oncology, British Columbia Cancer Agency, Vancouver, BC V5Z 4E6, Canada
* Correspondence: mehrnoosh.pauls1@bccancer.bc.ca (M.P.); schia@bccancer.bc.ca (S.C.)

**Abstract:** Breast cancer (BC) is the most common malignancy among women in Canada. Adjuvant treatment in early BC can reduce the risk of BC recurrence. Historically, the decision for adjuvant chemotherapy for early BC was made only based on clinical and tumour characteristics. In recent years, there has been an effort toward developing genomic assays as a predictive and prognostic tool to improve precision in estimating disease recurrence, sensitivity to systemic treatment and ultimately with clinical utility for guidance regarding adjuvant systemic treatment(s). There are various commercial genomic tests available for early-stage ER+/HER-2 negative BC. This paper will review the Oncotype DX 21-gene Recurrence Score (RS), MammaPrint, EndoPredict, Prosigna®, and Breast Cancer Index (BCI) genomic assays. We will also focus on these genomic assays' clinical application and utility in node-positive early-stage BC based on the most recent evidence and guidance recommendations.

**Keywords:** Oncotype DX 21-gene Recurrence Score (RS); MammaPrint; EndoPredict; Prosigna®; Breast Cancer Index (BCI)



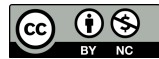

## 1. Introduction

Prior to genomic testing, the decision for adjuvant chemotherapy for early BC was made based on clinical and tumour characteristic results such as patient age, co-morbidities, the number of lymph nodes involved, tumour size and grade, hormonal biomarker (estrogen, progesterone), and human epidermal growth factor receptor 2 (HER-2) status [1]. However, there is a risk of overtreating BC patients if treatment is solely based on clinical and tumour characteristics. Therefore, genomic testing has become a source of interest in the past two decades, leading to the development of various genomic assays that are of prognostic and/or predictive significance to guide our treatment decisions in clinical setting. Several different genomic assays are currently approved and available, including the Oncotype DX, MammaPrint, EndoPredict, Prosigna®, and BCI [2]. This review paper will discuss each genomic assay, focusing on its clinical application and utility in node-positive early-stage BC based on the most recent prospective trial data.

## 2. Oncotype DX

The Oncotype DX 21-gene Recurrence Score (RS) is a first-generation genomic assay that uses reverse-transcriptase–polymerase-chain-reaction (RT–PCR) methodology initially validated to quantify the expression of 21 prospectively selected genes (both tumour associated and housekeeping) to provide an estimate of the risk of distance recurrence treated with adjuvant hormonal therapy alone [3]. This assay is the most validated assay among other assays providing both prognostic information regarding the risk of recurrence and predictive information to guide whether the use of adjuvant chemotherapy does further reduce the risk of recurrence of BC—as demonstrated in the node-negative prospective TAILORx clinical trial [4]. The utility of RS in node-positive disease was initially evaluated in a retrospective analysis trial that evaluated the prognostic and predictive value of RS

assay on tumours from postmenopausal women with node-positive, ER + BC from the SWOG-8814 phase 3 trial [5]. This trial demonstrated that RS is prognostic for disease-free survival (DFS) and overall survival (OS). The 10-year DFS was estimated as 60%, 49%, and 43% for low (score < 18), intermediate (score 18–30), and high-risk (score ≥ 31) group, respectively. The 10-year OS was estimated as 77%, 68%, and 51% for low, intermediate, and high-risk group, respectively. The RS assay was also predictive of the benefit of cyclophosphamide, doxorubicin, and fluorouracil (CAF) chemotherapy on DFS and OS. There was no apparent benefit in DFS and OS among patients for the addition of adjuvant CAF chemotherapy for RS scores less than 18 or 18–30. However, there was a clear benefit in both DFS (HR 0.59) and OS (HR 0.56) among patients when CAF chemotherapy was added to tamoxifen with a recurrence score of 31 and above [5].

This retrospective data led to the landmark prospective RxPONDER trial that evaluated RS's prognostic and predictive value among women with ER+/HER-2 negative, 1–3 positive axillary lymph node-positive (N1) BC with RS score of 25 or lower [6]. This was a phase 3 trial that included 5018 women of which 33.2% were premenopausal and 66.8% postmenopausal. In terms of choice of chemotherapy when chosen, 50% received docetaxel and cyclophosphamide, and 50% received anthracycline-based chemotherapy. This trial demonstrated no additional benefit of adjuvant chemotherapy to 5 years DFS in postmenopausal women with a recurrence score of 25 and lower regardless of number of nodes (1 versus 2–3), degree of nodal involvement (microscopic versus macroscopic) or range of RS (0–13 versus 14–25). The 5-year DFS was 91.9% among endocrine therapy alone vs. 91.3% among the chemo–endocrine group (HR 1.02). This ultimately supported safely omitting adjuvant chemotherapy that leads to overtreatment among postmenopausal women with ER+/HER-2 negative, with 1–3 positive nodes BC and RS scores of 25 or lower. However, this trial showed an improved DFS and distance relapse-free survival noted among pre-menopausal when adjuvant chemotherapy was added to endocrine therapy regardless of RS score or number of nodes involved (1 versus 2–3). Five years-DFS was 89.0% for the endocrine therapy alone group vs. 93.9% for the chemo–endocrine group (HR 0.58) among premenopausal women [6]. The preliminary OS presented at SABCS 2020 in abstract format also shows an improved OS (HR 0.47) among premenopausal women with RS ≤ 25 and 1–3 lymph nodes involvement that receive chemotherapy in addition to their endocrine therapy [7].

There however remains a lingering question of whether adjuvant chemotherapy adds benefits to that of optimal endocrine therapy of ovarian suppression and an aromatase inhibitor in this cohort of patients. Unfortunately, in the RxPONDER trial only 17% of pre-menopausal women randomized to endocrine therapy alone received ovarian suppression up to year 2 as their strategy of treatment. Therefore, there needs to be a prospective trial comparing ovarian suppression plus endocrine therapy vs. chemotherapy plus ovarian suppression and endocrine therapy among premenopausal women with ER+/HER-2 negative, 1–3 node-positive BC and RS scores of 25 or lower to address benefit from adjuvant chemotherapy in this endocrine optimally treated population.

Based on current evidence and the updated American Society of Clinical Oncology (ASCO) guidelines 2022, RS should be offered and guide decisions regarding adjuvant endocrine and chemotherapy among postmenopausal women with 1–3 node-positive ER+/HER-2 negative BC. In addition, chemo–endocrine therapy should be offered to the patients with RS score ≥ 26 (type: evidence-based; evidence quality: high; strength of recommendation: strong). RS should not be offered or used to guide systemic chemotherapy decisions among premenopausal women with 1–3 node-positive ER+/HER-2 negative BC (type: evidence-based; evidence quality: high; strength of recommendation: moderate). Currently, there is insufficient evidence for the use of RS for treatment guidance among patients with ≥4 node-positive BC (type: informal consensus; evidence quality: insufficient; Strength of recommendation: moderate) [8].

### 3. MammaPrint

70-gene assay (MammaPrint) is a genomic assay that initially used microarray analysis to evaluate the 70-gene prognosis profile and classify patients into high-risk and low-risk subgroups [9]. This genomic assay was initially developed by the Netherlands Cancer Institute, which developed a 70-gene expression assay from 78 frozen samples of lymph node-negative BC patients younger than 55 years of age at diagnosis and treated in the Netherlands Cancer Institute [10]. The same group of investigators later performed the first retrospective study to validate the prognostic significance of 70 gene signatures. In their study, they evaluated fresh frozen tumours from 295 patients with stage I or II BC a (144 lymph node-positive and 151 lymph node-negative) treated at the hospital of the Netherlands Cancer Institute with a successful ability to confirm that 70 gene signatures outperformed standard systems based on clinical and histologic criteria. Furthermore, they were able to successfully separate a group with an excellent prognosis at 10 years from a group with a high risk of recurrence before five years [9]. This study ultimately led to various other studies that further validate the prognostic role of the 70-gene assay both in node-negative and node-positive BC patients [11–13]. One advantage of the MammaPrint is the ability to use this assay across the intrinsic subtypes of breast cancer.

Microarray in Node-Negative and 1 to 3 Positive Lymph Node Disease May Avoid Chemotherapy (MINDACT trial) is one of the most prominent landmark prognostic trials that evaluates the value of a 70-gene assay and its feasibility for clinical application [14,15]. This trial was an international phase III prospective randomized study that enrolled 6693 women with early-stage BC that underwent both clinical and genomic risk assessment. Clinical risk assessment was evaluated using Adjuvant! Online and genomic risk assessment was evaluated using 70-gene signature (version 8.0 with HER2 status, www.adjuvantonline.com) to determine risk of breast cancer recurrence [14]. In this trial, 80% women had lymph node-negative and 20% of women had 1–3 positive lymph nodes early-stage breast cancer.

Patients were then divided into four groups base on their clinical and genomic risk:

1. Both high clinical and genomic risk women (27%);
2. Both low clinical and genomic risk women (41%);
3. Low clinical risk and high genomic risk women (8.8%);
4. High clinical risk and low genomic risk women (23.2%).

Women with both high clinical and genomic risk received chemotherapy, whereas women with low clinical and genomic risk omitted chemotherapy. Women with discordant clinical and genomic predictions were randomly assigned to receive or not receive adjuvant chemotherapy with the main focus to evaluate the 5-year distant metastases-free survival in this group [14].

Among women with low clinical risk and high genomic risk, no difference in the 5-year rate of survival without distant metastasis was identified with or without chemotherapy (95.0% versus 95.8%, respectively, *p* = 0.66). This finding suggests that the 70-gene panel cannot be used as a guidance tool to make decisions regarding systemic adjuvant chemotherapy decisions among patients with low clinical and high genomic risk. Interestingly, among women with high clinical risk and low genomic risk, 5-year distant metastasis-free survival rates were also identified with or without chemotherapy (95.9% versus 94.4%, respectively, *p* = 0.27) making a 70-gene panel a tool to guidance decision regarding systemic adjuvant chemotherapy and avoid overtreatment. Given these results, approximately 14% of women with BC may be spared adjuvant chemotherapy if one was to use the genomic strategy (i.e., MammaPrint) compared to the clinical strategy (Adjuvant! Online). It is important to note that these results were regardless of nodal status [14].

Recent long-term follow-up results from the MINDACT trial and exploratory analysis of the trial by age supported preliminary analysis confirming this trial as a positive de-escalation trial to avoid overtreating with chemotherapy. Updated 5-years distance metastasis-free survival (DMFS) among high clinical risk and low genomic risk BC patients that did not receive chemotherapy was 95.1% (95% CI 93.1–96.6), was non-inferior

to patients that received chemotherapy [16]. Long-term follow-up 8-year DMFS among high clinical risk and low genomic risk BC patients was 92% among patients that received chemotherapy vs. 89.4% among patients that did not get chemotherapy (HR 0.66, 95% CI 0.48–0.92). A further exploratory analysis evaluating the effect of age on these results showed the most benefit of adjuvant chemotherapy was among women aged 50 years and younger and nodal status did not make any changes to survival advantage from chemotherapy. The 8-year DMFS among women with high clinical risk and low genomic risk BC age 50 years and younger was 93.6% with chemotherapy vs. 88.6% without chemotherapy (absolute benefit of 5%, 95% CI 0.5–10.4), while the 8-year DMFS among women with high clinical risk and low genomic risk BC above age 50 years and older was 90.2% with chemotherapy vs. 90.0% without chemotherapy (absolute benefit of 0.2%, 95% CI −4.0–4.4) [16]. However, it is essential to note that exploratory analysis of the trial by age is underpowered, and a larger trial needs to investigate true benefit of adjuvant chemotherapy among women aged 50 and younger. Like the RxPONDER trial, there remains a lingering question of whether adjuvant chemotherapy adds benefits to optimal endocrine therapy of ovarian suppression and an aromatase inhibitor among women aged 50 and younger [6].

What is a significant difference in results between MINDACT and TAILORx is that high genomic grade by MammaPrint is not predictive of benefit of chemotherapy. In the low clinical risk/high genomic risk group, the patients randomized to adjuvant chemotherapy did not significantly benefit (DMFS HR 0·85 (0·53 to 1·37) [14–16]. Whereas in at least in the TAILORx (but not the RxPONDER trial) the higher the RS the greater the benefit to adjuvant chemotherapy (with RS $\geq$ 21) in pre-menopausal node-negative breast cancer [4]. Thus, it does not appear that MammaPrint has clinical utility for escalation of therapy.

Currently, based on the updated ASCO guidelines 2022, the MammaPrint test is recommended to guide adjuvant endocrine and chemotherapy decision among patients older than 50 with high clinical risk node-negative or 1–3 node-positive BC (type: evidence-based; evidence quality: intermediate; strength of recommendation: strong). MammaPrint test is not recommended to guide adjuvant treatment among women aged 50 or younger with high clinical risk node-negative or 1–3 node-positive BC (type: evidence-based; evidence quality: high; strength of recommendation: strong). The evidence for clinical utility of the MammaPrint test is insufficient for use among women with low-risk clinical BC regardless of age or menopause status (type: evidence-based; Evidence quality: intermediate; Strength of recommendation: moderate). Currently, there is insufficient evidence for the use of the MammaPrint test for treatment guidance among patients with $\geq$4 node-positive BC (type: informal consensus; evidence quality: insufficient; strength of recommendation: strong) [8].

### 4. Prosigna®

The Prosigna® is a genomic assay that measures the expression of 50 genes allowing classification into tumour subtypes. In addition, it provides a risk of recurrence score (ROR) that classifies patients into high-, medium-, and low-risk recurrence classes [17]. Prosigna® utilizes the NanoString nCounter and can be performed in local laboratories with the intent to be carried out in qualified routine hospital pathology laboratories [18]. Several retrospective studies included node-positive early-stage BC have demonstrated the clinical utility of PAM50 and ROR scores as a prognostic tool (ATAC trial, Austrian BC Study Group 8 (ABCSG-8), a Danish cohort study (DBCG)36 endocrine-treated women with early-stage BC, DBCG 77B) [19–22]. Initially, the prognostic significance of Prosigna® was clinically validated via assessment of mRNA from 1017 postmenopausal patients treated with adjuvant tamoxifen or anastrozole from an ATAC trial using the NanoString nCounter to predict the risk of recurrence [19]. In this analysis, ROR provided continuous prognostic information regarding the risk of distant recurrence at ten years in both node-negative and node-positive diseases. It is important to note that only 26% of the cohort had the node-positive disease in this analysis [19]. Further validation for PAM50 ROR as a prognostic tool came from the ABCSG-8 trial [20]. In this trial, 1478 hormone receptor-positive with early BC were treated with either five years of tamoxifen or two years of tamoxifen followed by

three years of anastrozole had their RNA extracted for PAM50 assessment. Both intrinsic subtype (luminal A/B, HER2-enriched, basal-like) and ROR scores were calculated to evaluate whether they added reliable prognostic value in predicting distant recurrence. Only 29% (431/1478) had the node-positive disease among women included in this study. In their study, they were able to successfully provide Level 1 evidence for clinical validity of the PAM50 test in predicting the risk of distance recurrence in postmenopausal women with ER+ early BC treated with adjuvant hormonal therapy [20]. The probability of 10-year distant relapse-free survival (DRFS) by subtyping was 93.9% for luminal A breast cancers and 82.2% for luminal B breast cancers, putting the luminal A cohort at a lower risk for relapse compared to the luminal B cohort (HR 2.85, *p*-value < 0.0001). The probability of 10-year distant relapse-free survival (DRFS) based on the ROR score cohort was 96.7% in the low-risk cohort, 91.3% in the intermediate cohort, and 79.9% in the high-risk cohort. These results were seen regardless of whether pathologic node involvement was present [20].

However, it is important to note that PAM50 and ROR scores are not predictive of benefits from treatment, such as length of adjuvant endocrine therapy or the need for adjuvant chemotherapy. No patient received adjuvant chemotherapy in the ABCSG-8 trial [20]. In addition, further analysis of women randomized to radiotherapy or control after breast conservation surgery also did not show predictive benefit of PAM50 testing for the benefit of adjuvant radiation [23]. However, this trial constitutes Level 1 evidence for clinical validity of the PAM50 test for predicting the risk of distance recurrence in postmenopausal women with ER+ early BC [20,23]. Long-term follow-up for a combination of ABCSG-8 trial and TransATAC study analysis further confirmed the validation of Prosigna® for predicting late distant recurrence and early distant recurrences [24]. Similar support for the use of Prosigna® as a prognostic tool came from a study led by the Danish BC Group [25].

There is a current large prospective trial (OPTIMA) evaluating the predictive capacity of Prosigna [ISRCTN 42400492]. The design effectively compares chemotherapy + endocrine therapy vs endocrine therapy alone for patients with low or intermediate Prosigna ROR scores in approximately 4500 patients. The study has a non-inferiority design with a co-primary end point of assessing the cost effectiveness evaluation of test-directed treatment.

Based on current evidence and updated ASCO guidelines 2022, the evidence is inconclusive to recommend using the Prosigna® test to guide decisions for adjuvant endocrine and chemotherapy among postmenopausal women with 1–3 node-positive BC (type: evidence-based; evidence quality: intermediate; strength of recommendation: moderate). Regardless of nodal status, the Prosigna® test is not recommended to guide decisions for adjuvant systemic chemotherapy among premenopausal women with BC (type: informal consensus; evidence quality: insufficient; strength of recommendation: moderate). There is currently insufficient evidence for the use of the Prosigna® test for treatment guidance among patients with ≥4 node-positive BC (Type: informal consensus; evidence quality: insufficient; strength of recommendation: strong) [8].

## 5. Endopredict

Endopredict (EP) is an RNA-based molecular test that assesses eight cancer-related and three reference genes to predict distant recurrence in ER+/Her-2 negative BC patients [26]. This genomic assay was initially presented and validated as a prognostic test by Filipits et al. in 2011, evaluating distance recurrence at 5 and 10 years among patients with ER+/HER-2 negative BC node-negative and node-positive disease treated with 5 years of endocrine therapy (ET) from two large randomized phase III trials (Austrian Breast and Colorectal Cancer Study Group (ABCSG)-6 and ABCSG-8) [26]. Their study also combined EP with clinical characteristics of nodal status and tumour size to generate a hybrid score named EPclin to improve the prognostic discrimination of the test in a clinical setting. They classified patients into a low-risk group (EPclin score < 3.3) and a high-risk group (EPclin score ≥ 3.3) and successfully demonstrated significantly different 10-year rates of distant recurrence rates between the low-risk group vs. high-risk group (4% low-risk EPclin score vs. 22–28% high-risk EPclin score respectively, *p* < 0.001), outperforming conventional

clinicopathological parameters [26]. Long term follow-up of the combined ABCSG-6/8 cohorts after 15 years continues to demonstrate the prognostic value of the EP and EPclin score in predicting early and late distant recurrence among women ER+/HER-2 negative BC [27]. It is important to note that EPclin scores were significant predictors of distant recurrence-free rate regardless of nodal status [27]. All cohorts from ABCSG-6 and ABCSG-8 were treated with endocrine therapy (ET) only [26]. EP and EPclin prognostic value for predicting disease recurrence has also been validated among ER+/HER-2 negative node-positive BC patients who received chemotherapy in addition to ET in phase III GEICAM 9906 [28]. In this trial, the rate of 10 years of metastasis-free survival in the low-risk EP score group was 93% vs. 70% in the high-risk EP score group (absolute risk reduction of 23%, $p < 0.0001$) and 100% for low-risk EPclin score group vs. 72% in the high-risk EPclin score group ($p < 0.0001$) [28]. In summary, all these studies were able to validate the prognostic implication of EP and EPclin among node-positive ER+/HER-2 negative BC whether they had only adjuvant ET or adjuvant chemotherapy followed by ET [26–28].

The predictive value of EPclin to determine the benefit from the addition of chemotherapy to ET has been investigated in a retrospective study estimating 10-year distant recurrence-free interval (DRFI) rates among patients with ER+/HER-2 negative BC who received adjuvant endocrine therapy (ET) alone compared to those with chemotherapy plus endocrine therapy (ET + C) from pooled population in five large clinical trials (ABCSG-6, ABCSG-8, and TransATAC trials received five years of ET only versus GEICAM/9906 and GEICAM 2003/02 trials received ET + C) [29]. This study showed no significant difference in 10-year DRFI among EPclin scores for the low-risk group whether they received ET alone versus ET + C, thus, identifying patients who may have been over treated with chemotherapy. However, among the EPclin scores in the high-risk group, there was a difference in 10-year DRFI between patients who received ET alone versus ET + C leading to the identification of patients who benefit from the addition of chemotherapy (eg., EPclin score of 5 had a 10-year DR risk of 46% compared to 26% for women who received ET + C, an absolute risk difference of 20%). This retrospective study demonstrated that a high EPclin score can predict chemotherapy benefits among women with ER+/HER2-negative BC [29].

Currently, based on the updated ASCO guidelines 2022, clinicians can use the EndoPredict test to guide decisions for adjuvant endocrine and chemotherapy among post-menopausal women with BC that is node-negative or 1–3 node-positive (type: evidence-based; evidence quality: intermediate; strength of recommendation: moderate). However, the EndoPredict test is not recommended to guide adjuvant endocrine and chemotherapy treatment decisions among premenopausal women with node-negative or node-positive BC (type: informal consensus; evidence quality: insufficient; strength of recommendation: moderate). Currently, there is insufficient evidence for the use of the EndoPredict test for treatment guidance among patients with ≥4 node-positive BC (Type: evidence-based; Evidence quality: intermediate; Strength of recommendation: moderate) [8].

## 6. Breast Cancer Index (BCI)

BCI (HOXB13/IL17BR ratio (H/I)) is a genomic assay that utilizes the RT–qPCR assay to analyze the expression of seven genes and classifies BC patients into low, intermediate, and high-risk groups [30]. Initially, BCI (H/I) genomic assay's prognostic performance for predicting early and late BC recurrence was validated via retrospective analyses of samples from node-negative early ER+ BC from two prospective trials in Stockholm and a multi-institutional cohort [31]. Further studies also have looked at the prognostic performance of BCI (H/I) genomic assay for predicting distance recurrence in 1–3 node-positive early ER + BC [32]. In a study carried out by Zhang et al., BCI (H/I) gnomic assay was able to stratify 1–3 node-positive early ER + BC patients into low-risk and high-risk groups and show a significantly lower rate of distance recurrence among BCI-low risk group (15-years risk of distance recurrence of 1.3% among BCI-low risk group versus 29% among BCI-high risk group, HR 25.93, $p < 0.0001$) [32]. This study was successfully able to show prognostic utility of BCI (H/I) among early-stage 1–3 node-positive BC [32].

The predictive component of BCI (H/I) to assess benefit from extended endocrine therapy has been validated in multiple different studies (National Cancer Institute of Canada Clinical Trials Group MA.17 trial and trans-adjuvant Tamoxifen-To Offer More (Trans-aTTom) trial) [33,34]. MA. 17 trial was a prospective randomized phase III trial that evaluated the role of letrozole as extended adjuvant therapy among postmenopausal women with ER + BC ($n$ = 5157) that completed standard five years of adjuvant tamoxifen and remained disease-free. In the correlative study performed retrospectively, tumour blocks from MA.17 were tested for BCI (H/I) genomic assay to predict late recurrence and benefit from extended endocrine therapy. In their study, they demonstrated that BCI (H/I) genomic testing was successfully able to identify patients who have benefited from extended endocrine therapy. Patients with high BCI (H/I) had a 5-years recurrence-free survival (RFS) of 73% among the placebo group versus 89.5% among extended endocrine therapy with the letrozole group (absolute risk reduction of 16.5%, $p$ = 0.007), thus, demonstrating that patients with high BCI (H/I) would benefit from the addition of extended endocrine therapy. However, patients with low BCI (H/I) did not have a significant reduction in their 5-year RFS (absolute risk reduction of 4%, $p$ = 0.35) and therefore may not benefit from extended endocrine therapy with letrozole. The interaction test was positive suggestive of predictive significance for the genomic assay [33]. Data from Trans-aTTom trial further support results from MA.17 trial. In the Trans-aTTom trial, high BCI (H/I) patients also derived benefits from 10-years versus 5-years of adjuvant tamoxifen (HR 0.35, 10.2% absolute benefit, $p$ = 0.027). In contrast, low BCI (H/I) patients did not have significant benefits from extended endocrine therapy (HR 1.01, $p$ = 0.768) [34].

As a result, based on current evidence and the updated ASCO guidelines 2022, the BCI (H/I) test can be offered to guide decisions about extended endocrine therapy among patients with node-negative or 1–3 node-positive BC who have been treated with five years of primary endocrine therapy without evidence of recurrence (type: evidence-based; evidence quality: intermediate; strength of recommendation: moderate). Currently, there is insufficient evidence for the use of the BCI (H/I) test for treatment guidance among patients with ≥4 node-positive BC (type: evidence-based; evidence quality: intermediate; strength of recommendation: moderate) [8].

## 7. Conclusions

In conclusion, various genomic assays have demonstrated the ability to provide independent prognostic value and outperform standard clinicopathological criteria as a prognostic tool. Although each genomic assay looks at different gene sets and has its separate platform, most can successfully identify a low-risk group of BC and provide prognostic information regarding distance recurrence. However, not all genomic assays have validated evidence for their predictive performance, which ultimately provides greater clinical utility to guide treatment decisions and reduce overtreatment. Based on evidence quality and the 2022 updated ASCO guidelines, the Oncotype DX and MammaPrint genomic tests are recommended to guide adjuvant endocrine and chemotherapy decisions among postmenopausal women with 1–3 node-positive ER+/Her-2 negative BC (strong recommendation) [8]. The evidence behind this recommendation mostly comes from RxPONDER trial and MINDACT trial [6,14]. However, there is insufficient evidence to recommend any genomic testing in premenopausal with 1–3 node-positive ER+/HER-2 negative BC to guide adjuvant treatment (Figure 1). Therefore, further work is needed to evaluate the role of genomic testing among premenopausal women with BC, particularly in the face of optimal endocrine therapy. The evidence is inconclusive to recommend using the Prosigna® test to guide decisions for adjuvant therapy in node-positive BC base on ASCO guideline 2022 [8]. We do have to wait for results of OPTIMA trial evaluating the predictive capacity of Prosigna. Finally, BCI (H/I) testing has been validated more regarding its predictive ability to guide decisions about extended endocrine therapy and not chemotherapy. As a result, ASCO guidelines currently recommend use of BCI (H/I)

testing to guide decisions regarding only extended endocrine therapy for women with ER+, node-positive BC [8].

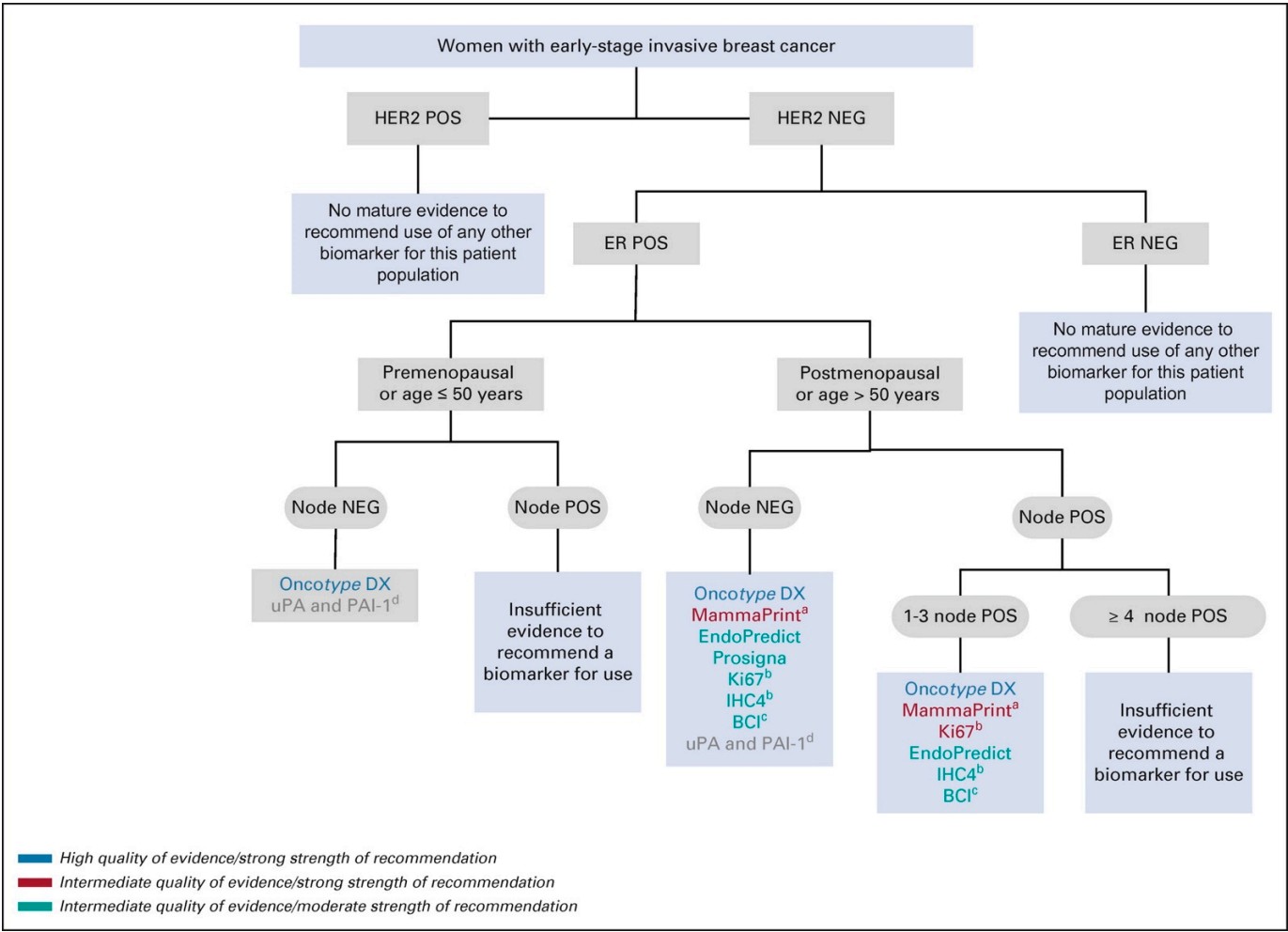

**Figure 1.** Algorithm on usage of biomarkers to guide decisions on adjuvant treatment among patients with node-positive early-stage ER+/HER2– BC as per ASCO updated guidelines 2022. Reprinted with permission from ASCO guidelines. Copyright year 2022, copyright owner's Wolters Kluwer Health, INC [8].

Current clinical practice should be aligned with the current ASCO guidelines for using Oncotype Dx and MammaPrint to guide decisions regarding adjuvant endocrine and chemotherapy among postmenopausal women diagnosed with ER+/Her-2 negative 1–3 node-positive early-stage breast cancers since both genomic testing have the most robust evidence providing important prognostic information regarding the de-escalation of adjuvant chemotherapy. Although ASCO guidelines state that EndoPredict can be used as a predictive genomic tool to guide decisions regarding adjuvant endocrine and chemotherapy, we believe in the absence of a sufficiently powered prospective randomized clinical trial, the two above described genomic assays provide more robust data to provide clinicians and patient confidence in the de-escalation of therapy. Similarly, we will need to wait for prospective level 1 evidence for Prosigna® genomic testing as a predictive tool before it is commonly incorporated into clinical practice.

In summary, the evidence presented in this review supports the use of genomic testing to guide clinical decision making in early-stage ER+/EHR2 breast cancer (Table 1). The result of this is a large step forward to personalized medicine for appropriate individuals in our clinics.

**Table 1.** Summary table of all the molecular assays discussed in review paper and significant trail associated with each assay.

| Assay | Platform | Number of Gene Selected | Pivotal Studies Associated with the Assay for 1–3 Node-Positive BC | Recommendation by ASCO Clinical Practice Guideline for Postmenopausal Women with 1–3 Node-Positive ER+/HER-2 BC | Recommendation by ASCO Clinical Practice Guideline for Premenopausal Women with 1–3 Node-Positive ER+/HER-2 Negative BC |
|---|---|---|---|---|---|
| Oncotype DX | RT–PCR | 21 genes | SWOG-8814 (retrospective trial) RxPONDER trial (prospective trial) | Oncotype DX should be offered or used to guide systemic endocrine and chemotherapy decisions Evidence quality: **high:** Strength of recommendation: **strong** | Oncotype Dx should **NOT** be offered or used to guide systemic endocrine and chemotherapy decisions Evidence quality: **high;** Strength of recommendation: **moderate** |
| MammaPrint | Microarray analysis | 70-gene | MINDACT trial (prospective trial) | MammaPrint should be offered to guide adjuvant endocrine and chemotherapy decision among patients older than 50 with high clinical riskEvidence quality: **intermediate**; Strength of recommendation: **strong** | MammaPrint test is **NOT** recommended to guide adjuvant treatment among women age 50 or younger with high clinical risk Evidence quality: **high** -Strength of recommendation: **strong**. |
| Prosigna® | NanoString nCounter | 50 genes | ATAC trial, ABCSG A Danish cohort study DBCG 36 endocrine-treated women with early-stage BC DBCG 77B (all retrospective trials) OPTIMA (prospective trial)-awaiting results | Inconclusive to recommend using the Prosigna® test to guide decisions for adjuvant endocrine and chemotherapy Evidence quality: **intermediate**; Strength of recommendation: **moderate** | Prosigna® test is **NOT** recommended to guide decisions for adjuvant systemic endocrine and chemotherapy Evidence quality: **insufficient**; Strength of recommendation: **moderate** |
| Endopredict | RT–PCR | 8 genes | Predictive value of Endopredict is from pooled population from five large retrospective clinical trials (ABCSG-6, ABCSG-8, TransATAC trials, GEICAM 9906, GEICAM 2003/02) | EndoPredict can be used by clinicians to guide decisions for adjuvant endocrine and chemotherapy Evidence quality: **intermediate**; Strength of recommendation: **moderate** | EndoPredict test is **NOT** recommended to guide adjuvant endocrine and chemotherapy Evidence quality: **insufficient**; Strength of recommendation: **moderate** |
| Breast Cancer Index (BCI) | RT–PCR | 7 genes | MA.17 trial Trans-aTTom trial) | BCI test can be offered to guide decisions **ONLY regarding** extended endocrine therapy among patients with node-negative or 1–3 node-positive BC who have been treated with five years of primary endocrine therapy without evidence of recurrence Evidence quality: **intermediate**; Strength of recommendation: **moderate** | BCI test can be offered to guide decisions **ONLY regarding** extended endocrine therapy among patients with node-negative or 1–3 node-positive BC who have been treated with five years of primary endocrine therapy without evidence of recurrence Evidence quality: **intermediate**; Strength of recommendation: **moderate** |

**Author Contributions:** M.P. and S.C. contributed to the conception, literature review, drafting of the manuscript, and final submission. All authors have read and agreed to the published version of the manuscript.

**Funding:** This review paper did not receive any funding.

**Acknowledgments:** The authors thank the current oncology journal for the opportunity to participate in the evolving paradigm of curative intent BC management special edition.

**Conflicts of Interest:** Dr. Pauls receives a consulting honoraria and sponsorship from Ipsen, Novartis and Pfizer. All outside of the submitted work. Dr Chia reports honoraria from Novartis, Hoffmann LaRoche, Pfizer, Eli Lilly, Merck, AstraZeneca, Exact Sciences and Gilead.

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
