# Peer review of "Clinical Utility of Genomic Assay in Node-Positive Early-Stage Breast Cancer"

_curroncol, doi:10.3390/curroncol29070407_

Round 1

Reviewer 1 Report

Dear Authors,

Congratulations on a strong review of the current evidence for genomic assays in node-positive early-stage breast cancer.  We had the following minor suggestions to your manuscript:

1) Could you please include a summary table comparing the characteristics of the molecular assays and summarizing the significant trails associated with each assay for the readers. 

2) Your review of the MINDACT trial for MammaPrint should include a discussion of the following manuscript: Piccart et al. 70-gene signature as an aid for treatment decisions in early breast cancer: updated results of the phase 3 randomised MINDACT trial with an exploratory analysis by age. Lancet Oncol. 2021 Apr;22(4):476-488.  

3) In regard to the recent ASCO guideline recommendations for Oncotype Dx and MammaPrint, a recommendations was made supporting the use of both assays to guide decisions regarding adjuvant endocrine and chemotherapy among postmenopausal women or age 50 or older with node-negative or 1-3 node positive early-stage breast cancer.  Could you please clarify for the readers the reasoning for the Evidence Quality Rating Strong versus Intermediate when comparing Oncotype Dx and MammaPrint.  Both assays are supported with large prospective randomized clinical trials yielding similar results.  Do you agree with this rating and could your expand on the difference in evidence quality?

4) In your review of EndoPredict, you may wish to note that the predictive evidence for using this assay is generated from a non-randomized, retrospective pooled analysis from previous clinical trials. [reference 28: Sestak et al. Breast Cancer Research and Treatment).  In your opinion, do you think the ASCO recommendation that clinicians may use EndoPredict to guide decisions for adjuvant endocrine and chemotherapy among postmenopausal women with 1-3 node positive early-stage breast cancer was warranted based this single retrospective review as opposed to an inconclusive recommendation?  

5) I would have liked the manuscripts conclusion to expand further on the authors own insight as to the strengths and weaknesses of the recent ASCO clinical guideline recommendation if possible. [Reference 8: Andre et al.  Journal of Clinical Oncology)

6) In Figure 1, under 1-3 node positive, the Breast Cancer Index (BCI) recommendation applies to clinical decision making for extended adjuvant endocrine therapy.  This should be clarified in a footnote below the figure.     

Author Response

Dear Current Oncology reviewer

Thank you for your review and response to our manuscript entitled " Clinical Utility of Genomic Assay in Node-Positive Early-Stage Breast Cancer ". Your comments were much appreciated. Please see below a response to the comments provided by the reviewers, detailing all changes that were made to the manuscript.

Please see the revised manuscript and response to the reviewer’s comments. We hope that these changes will improve the manuscript and make it a valuable contribution to your journal. We look forward to your response.

Sincerely,

Dr. Mehrnoosh Pauls & Dr. Stephen Chia 

BC Cancer Vancouver Centre, University of British Columbia

600 W 10th Ave

Vancouver, BC V5Z 4E6.

Reviewer(s)' Comments to Author:

We had the following minor suggestions for your manuscript:

1) Could you please include a summary table comparing the characteristics of the molecular assays and summarizing the significant trails associated with each assay for the readers. 

Author Response: We have inserted Table 3 with a summary of all the molecular assays discussed in this paper and the significant trail associated with each assay. Please refer below.

2) Your review of the MINDACT trial for MammaPrint should include a discussion of the following manuscript: Piccart et al. 70-gene signature as an aid for treatment decisions in early breast cancer: updated results of the phase 3 randomised MINDACT trial with an exploratory analysis by age. Lancet Oncol. 2021 Apr;22(4):476-488.  

Author Response: We have added the following paper summary and citation.

Recent long-term follow-up results from the MINDACT trial and exploratory analysis of the trial by age supported preliminary analysis confirming this trial as a positive de-escalation trial to avoid overtreating with chemotherapy. Updated 5-years distance metastasis-free survival (DMFS) among high clinical risk and low genomic risk BC patients that did not receive chemotherapy was 95.1% (95% CI 93.1-96.6), was non-inferior to patients that received chemotherapy. Long-term follow-up 8-years DMFS among high clinical risk and low genomic risk BC patients was 92% among patients that received chemotherapy vs. 89.4% among patients that did not get chemotherapy (HR 0.66, 95% CI 0.48-0.92). A further exploratory analysis evaluating the effect of age on these results showed the most benefit of adjuvant chemotherapy was among women aged 50 years and younger and nodal status did not make any changes to survival advantage from chemotherapy. The 8-year DMFS among women with high clinical risk and low genomic risk BC age 50 years and younger was 93.6% with chemotherapy vs. 88.6% without chemotherapy (absolute benefit of 5%, 95% CI 0.5-10.4). While the 8-year DMFS among women with high clinical risk and low genomic risk BC above age 50 years and older was 90.2% with chemotherapy vs. 90.0% without chemotherapy (absolute benefit of 0.2%, 95% CI -4.0-4.4). However, it is essential to note that exploratory analysis of the trial by age is underpowered, and a larger trial needs to investigate the true benefit of adjuvant chemotherapy among women aged 50 and younger. Like the RxPONDER trial, there remains a lingering question of whether adjuvant chemotherapy adds benefits to optimal endocrine therapy of ovarian suppression and an aromatase inhibitor among women aged 50 and younger.

What is a significant difference in results between MINDACT and TAILORx is that a high genomic grade by MammaPrint is not predictive of the benefit of chemotherapy. In the low clinical risk/high genomic risk group – the patients randomized to adjuvant chemotherapy did not significantly benefit (DMFS HR 0·85 (0·53 to 1·37)). Whereas in at least in the TAILORx (but not the RxPONDER trial) the higher the RS the greater the benefit to adjuvant chemotherapy (with RS ≥21) in pre-menopausal node-negative breast cancer. Thus it does not appear MammaPrint has clinical utility for escalation of therapy.

3) In regard to the recent ASCO guideline recommendations for Oncotype Dx and MammaPrint, a recommendation was made supporting the use of both assays to guide decisions regarding adjuvant endocrine and chemotherapy among postmenopausal women or age 50 or older with node-negative or 1-3 node-positive early-stage breast cancer.  Could you please clarify for the readers the reasoning for the Evidence Quality Rating Strong versus Intermediate when comparing Oncotype Dx and MammaPrint.  Both assays are supported with large prospective randomized clinical trials yielding similar results.  Do you agree with this rating and could you expand on the difference in evidence quality?

Author Response: This is an excellent point. Both trials were large prospective randomized clinical trials that led to similar conclusions in the post-menopausal cohort. Both Oncotype Dx and MammaPrint are helpful tools to guide decisions regarding adjuvant endocrine and chemotherapy among postmenopausal women or age 50 or older with node-negative or 1-3 node-positive early-stage breast cancer. One reason for MammaPrint being ranked as intermediate evidence quality rating instead of strong is likely because the exploratory analysis of the trial by age was underpowered base on long-term follow-up paper. Lancet Oncol. 2021 Apr;22(4):476-488. Furthermore, there are two prospective RCT with Oncotype DX relative to the one with MammaPrint.

4) In your review of EndoPredict, you may wish to note that the predictive evidence for using this assay is generated from a non-randomized, retrospective pooled analysis from previous clinical trials. [reference 28: Sestak et al. Breast Cancer Research and Treatment).  In your opinion, do you think the ASCO recommendation that clinicians may use EndoPredict to guide decisions for adjuvant endocrine and chemotherapy among postmenopausal women with 1-3 node-positive early-stage breast cancer was warranted based on this single retrospective review as opposed to an inconclusive recommendation?  

Author Response: We agree that based on current evidence, Oncotype Dx and MammaPrint have the strongest evidence to guide decisions regarding adjuvant endocrine and chemotherapy among postmenopausal women. EndoPredict is not a predictive tool since, as you indicated, the evidence is not strong and is based on one non-randomized, retrospective pooled analysis. In addition, there needs to be a large prospective randomized clinical trial to strengthen the use of EndoPredict as a predictive tool. Given that there is still retrospective predictive evidence for EndoPredict, ASCO guidelines use the wording of “can be used” instead of inconclusive recommendation. However, we agree with your statement, and clinically, EndoPredict is not currently used routinely as a predictive tool based on the current evidence. 

5) I would have liked the manuscript's conclusion to expand further on the author's own insight as to the strengths and weaknesses of the recent ASCO clinical guideline recommendation if possible. [Reference 8: Andre et al.  Journal of Clinical Oncology)

Author Response: We have added the following statement below in the conclusion section.

Current clinical practice should be aligned with the current ASCO guidelines for using Oncotype Dx and MammaPrint to guide decisions regarding adjuvant endocrine and chemotherapy among postmenopausal women diagnosed with ER+/Her-2 negative 1-3 node-positive early-stage breast cancers since both genomic testings have the most robust evidence providing important prognostic information regarding de-escalation of adjuvant chemotherapy. Although ASCO guidelines state that EndoPredict can be used as a predictive genomic tool to guide decisions regarding adjuvant endocrine and chemotherapy, we believe in the absence of a sufficiently powered prospective randomized clinical trial, the two above-described genomic assays provide more robust data to provide clinicians and patients confidence in de-escalation of therapy. Similarly, we will need to wait for prospective level 1 evidence for Prosigna® genomic testing as a predictive tool before it is commonly incorporated in clinical practice.

6) In Figure 1, under 1-3 node-positive, the Breast Cancer Index (BCI) recommendation applies to clinical decision-making for extended adjuvant endocrine therapy.  This should be clarified in a footnote below the figure. 

Author Response: We have added a footnote below the figure to clarify.

Table 3: Summary table of all the molecular assays discussed in the review paper and significant trail associated with each assay.

Assay

Platform

Number of Gene Selected

Pivotal Studies Associated with the Assay for 1-3 Node positive BC

Recommendation by ASCO clinical practice Guideline for Postmenopausal women with 1-3 node-positive ER+/HER-2 BC

Recommendation by ASCO clinical practice Guideline for Premenopausal women with 1-3 node-positive ER+/HER-2 negative BC

Oncotype DX

RT-PCR

21  genes

-SWOG-8814

(retrospective trial)

-RxPONDER trial (prospective trial)

Oncotype DX should be offered or used to guide systemic endocrine & chemotherapy decisions

-Evidence quality: high

-Strength of recommendation: strong

Oncotype Dx should NOT be offered or used to guide systemic endocrine & chemotherapy decisions

-Evidence quality: high

-Strength of recommendation: moderate

MammaPrint

Microarray analysis

70-gene

-MINDACT trial (prospective trial)

MammaPrint should be offered to guide adjuvant endocrine & chemotherapy decision among patients older than 50 with high clinical risk

-Evidence quality: intermediate -Strength of recommendation: strong

MammaPrint test is NOT recommended to guide adjuvant treatment among women age 50 or younger with high clinical risk

-Evidence quality: high

-Strength of recommendation: strong.

Prosigna®

NanoString nCounter

50 genes

-ATAC trial,

-ABCSG

-A Danish cohort study DBCG 36 endocrine-treated women with early-stage BC

-DBCG 77B (all retrospective trials)

-OPTIMA (prospective trial)-awaiting results

Inconclusive to recommend using the Prosigna® test to guide decisions for adjuvant endocrine and chemotherapy

-Evidence quality: intermediate -Strength of recommendation: moderate

Prosigna® test is NOT recommended to guide decisions for adjuvant systemic endocrine and chemotherapy

-Evidence quality: insufficient

-Strength of recommendation: moderate

Endopredict

RT-PCR

8 genes

Predictive value of Endopredict is from pooled population from five large retrospective clinical trials (ABCSG-6, ABCSG-8, TransATAC trials, GEICAM 9906, GEICAM 2003/02)

EndoPredict can be used by clinicians to guide decisions for adjuvant endocrine & chemotherapy

-Evidence quality: intermediate

-Strength of recommendation: moderate

EndoPredict test is NOT recommended to guide adjuvant endocrine & chemotherapy

-Evidence quality: insufficient

-Strength of recommendation: moderate

Breast Cancer Index (BCI)

RT-PCR

7 genes

MA.17 trial

Trans-aTTom trial)

BCI test can be offered to guide decisions ONLY regarding extended endocrine therapy among patients with node-negative or 1-3 node-positive BC that have been treated with five years of primary endocrine therapy without evidence of recurrence

-Evidence quality: intermediate

-Strength of recommendation: moderate

BCI test can be offered to guide decisions  ONLY regarding extended endocrine therapy among patients with node-negative or 1-3 node-positive BC that have been treated with five years of primary endocrine therapy without evidence of recurrence

-Evidence quality: intermediate

-Strength of recommendation: moderate

Thank you for taking your time for great feedback with this review paper.

M.Pauls

Reviewer 2 Report

This is a nice summary of currently available genomic assays for early stage ER/PR+ HER2- breast cancer. I only have the following minor comments:

- Change "commercially" to "commercial" (page 1, line 13).

- Define ASCO at first use (page 2, line 85).

- The paper may benefit from another table summarizing and comparing the genomic assays discussed.

Author Response

Dear Current Oncology reviewer

Thank you for your review and response to our manuscript entitled " Clinical Utility of Genomic Assay in Node-Positive Early-Stage Breast Cancer ". Your comments were much appreciated. Please see below a response to the comments provided by the reviewers, detailing all changes that were made to the manuscript.

Please see the revised manuscript and response to the reviewer’s comments. We hope that these changes will improve the manuscript and make it a valuable contribution to your journal. We look forward to your response.

Sincerely,

Dr. Mehrnoosh Pauls & Dr. Stephen Chia 

BC Cancer Vancouver Centre, University of British Columbia

600 W 10th Ave

Vancouver, BC V5Z 4E6.

Reviewer(s)' Comments to Author:

This is a nice summary of currently available genomic assays for early-stage ER/PR+ HER2- breast cancer. I only have the following minor comments:

  1. Change "commercially" to "commercial" (page 1, line 13).
  2. Define ASCO at first use (page 2, line 85).
  3. The paper may benefit from another table summarizing and comparing the genomic assays discussed.

Author Response:

1. We have changed "commercially" to "commercial"

  1. We have added the definition of the American Society of Clinical Oncology (ASCO) in the first use.

  2. We have inserted Table 3 with a summary of all the molecular assays discussed in this paper and the significant trail associated with each assay (please refer to the attachment table). 
